# ColdExpand: Semi-Supervised Graph Learning in Cold Start

## Abstract

Most real-world graphs are dynamic and eventually face the cold start problem. A fundamental question is how the new cold nodes acquire initial information in order to be adapted into the existing graph. Here we postulates the cold start problem as a fundamental issue in graph learning and propose a new learning setting, "Expanded Semi-supervised Learning." In expanded semi-supervised learning we extend the original semi-supervised learning setting even to new cold nodes that are disconnected from the graph. To this end, we propose ColdExpand model that classifies the cold nodes based on link prediction with multiple goals to tackle. We experimentally prove that by adding additional goal to existing link prediction method, our method outperforms the baseline in both expanded semi-supervised link prediction (at most 24%) and node classification tasks (at most 15%). To the best of our knowledge this is the first study to address expansion of semi-supervised learning to unseen nodes.

## 1 Introduction

Graph-based semi-supervised learning has attracted much attention thanks to its applicability to real-world problems. For example, a social network is graph-structured data in which people in the network are considered to be nodes and relationships between people are considered to be edges: two people are friends or sharing posts, etc. With this structural information, we can infer some unknown attributes of a person (node) based on the information of people he is connected to (i.e., *semi-supervised node classification*). In the case of retail applications, customers and products can be viewed as heterogeneous nodes and edges between customers and products can represent relationships between the customers and the purchased products. Such a graph can be used to represent spending habits of each customer and we can recommend a product to a user by inferring the likelihood of connection between the user and the product (i.e., *semi-supervised link prediction*).

Recent progress on Graph Neural Networks (GNN) (Bruna et al., 2013; Kipf & Welling, 2016a; Gilmer et al., 2017; Veličković et al., 2018; Jia et al., 2019) allows us to effectively utilize the expressive power of the graph-structured data and to solve various graph related tasks. Early GNN methods tackled semi-supervised node classification task, a task to label all nodes within the graph when only a small subset of nodes is labeled, achieving a satisfactory performance (Zhou et al., 2004). Link prediction is another graph-related task that was covered comparatively less than other tasks in the field of GNNs. In the link prediction task, the goal is to estimate the likelihood of connection between two nodes given node feature data and topological structure data. Link prediction can be used in recommendation tasks (Chen et al., 2005; Sarwar et al., 2001; Lika et al., 2014; Li & Chen, 2013; Berg et al., 2017) or graph completion tasks (Kazemi & Poole, 2018; Zhang & Chen, 2018). Most of the work on semi-supervised graph learning and link prediction assumes a static graph, that is, the structural information is at least "partially" observable in terms of nodes.

In the real world, however, new users or items can be added (as nodes) without any topological information (Gope & Jain, 2017). This is also referred as the *cold start problem* when a new node is presented to an existing graph without a single edge. In contrast to the warm start case, in which at least some topological information is provided, the cold start problem is an extreme case where there isn't any topological information to refer. In this setting, previous semi-supervised learning algorithms can not propagate information to the cold nodes. Even though the cold start problem is an extreme setting, it is an inevitable problem that occurs often in the real world data.

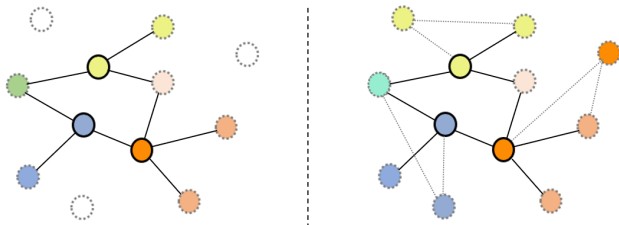

Figure 1: Left: Original Semi-supervised Learning. Right: Expanded Semi-supervised Learning. Nodes with solid lines are labeled nodes and other nodes with dot lines are unlabeled nodes. Expanded semi-supervised learning extends the original semi-supervised learning even to new nodes that are disconnected from the graph. By adding unseen cold nodes to the training we can not only expand learning to cold nodes but also use node features of cold nodes to better approximate the marginal distribution of the original graph data.

In this paper, we postulates the cold start problem as a fundamental issue in graph learning and propose a new learning setting, expanded semi-supervised learning. In expanded semi-supervised learning we extend the original semi-supervised learning setting even to new cold nodes that are disconnected from the graph as shown in Figure 1. To this end, we suggest the ColdExpand method, the method that uses multi-task learning strategy to alleviate the cold start problem that typical graph learning methods face. We experimentally prove that by adding an additional goal to the existing link prediction method, performance on the link prediction task is enhanced in every benchmark dataset and at most by 24%. We also prove that our method can expand semi-supervised node classification even to unseen, cold nodes. To the best of our knowledge, this is the first study to expand semi-supervised learning methods to unseen nodes. In the next section, we briefly introduce related works. In Section 3, we define our problem definition. Finally, in Section 4 we propose our ColdExpand model followed by our experimental environments along with corresponding results.

## 2 RELATED WORK

### 2.1 SEMI-SUPERVISED NODE CLASSIFICATION

Semi-supervised learning is a combination of unsupervised learning and supervised learning where true labels are only partially given. Semi-supervised learning algorithms aim to improve the model's performance by using unlabeled data points on top of labeled ones to approximate the underlying marginal data distribution (Van Engelen & Hoos, 2020). One of the most popular fields of semi-supervised learning is graph based semi-supervised node classification where all nodes should be classified when only a few node labels are given. In order to smooth the given subset of labels throughout the graph, various methods have been studied to effectively represent nodes. Deep Walk (Perozzi et al., 2014), LINE (Tang et al., 2015), node2vec (Grover & Leskovec, 2016) were early deep-learning-based methods targeting the node classification task by learning latent representations from truncated random walks. However, these models fail to share parameters between nodes causing learning to be inefficient. Kipf & Welling (2016a) introduced Graph Convolutional Networks (GCN) which use an efficient layer-wise propagation rule by approximating the first-order of spectral graph convolution. By limiting the spectral convolution to the first-order, GCNs not only lightened computation cost of the operation but also alleviated the over-fitting problem that previous spectral methods had. Commonly, GCN model is formed from stacked convolutional layers; the number of which decides how many hops of neighbors to consider for the convolution operation. Gilmer et al. (2017) presented Message Passing Neural Networks (MPNN) as a general form of spatial convolution operation and treated GCN as specific kind of a message passing process. In MPNNs, information between nodes is delivered directly by edges without visiting any spectral domains.

## 2.2 LINK PREDICTION

Links in graph-structured data appeal important interactions between nodes that may not be represented solely by node attributes. To this end, link prediction is an important task in order to effectively utilize graph data. In the link prediction task, we aim to predict whether an edge exists between two nodes from the same graph. Predicting the existence of links between nodes is useful for many applications including recommendation systems and graph completion tasks.

On a high level, link prediction methods can be separated into similarity-based and learning-based methods. Similarity-based link prediction is the simplest way to predict the link by assigning a similarity score $S_{xy}$ between two nodes $x$, $y$. Common Neighbors (Marchette & Priebe, 2008), Salton Index, Jaccard Index (Leydesdorff, 2008), Sorensen Index, Hub Promoted Index, Hub Depressed Index (Zhou et al., 2009), Leicht-Holme-Newman Index (LHN1) (Leicht et al., 2006) each define similarity scores by using the overlapping set of neighboring nodes between two nodes. While the former neighborhood-based methods focus on somewhat local similarity, other similarity-based methods consider global similarity by taking into account higher order links. Katz Index (Katz, 1953) considers the ensemble of all paths between two nodes summed with quadratic weight according to the distance of each path. SimRank (Jeh & Widom, 2002) also uses random walk process but measures how soon two random walkers from the two different nodes meet. However, when a cold node is introduced, the required structural information needed for previous similarity-based approaches is missing, thus making them impossible to be applied in cold start situations (Wang et al., 2016).

On the other hand, learning-based methods formulate link prediction as a binary classification problem given feature embedding vectors of two nodes; if an edge exists between the two nodes a positive label is given and vice versa. Kipf & Welling (2016b) introduced the graph auto-encoder (GAE), an unsupervised learning technique, which aims to reconstruct the original adjacency matrix. As a variant of GAE, variational graph auto-encoders (VGAE) additionally train a separate GCN module resulting in the logarithm of the variant matrix, $\log(\sigma^2)$. Additional Kullback-Leibler divergence loss between the distribution of encoder and decoder is added to the original reconstruction loss. Pan et al. (2019) further improved upon GAEs by applying adversarial training to regularize the latent node embedding matrix $Z$ to match a prior distribution of the original node feature matrix $X$. A simple MLP-based discriminator $D$ was used for adversarial learning between the real distribution of $G = \{X, A\}$ and the latent node matrix $Z$. Unlike similarity-based methods, learning-based methods can be applied to cold start problems since they make use of node feature information along with topological structure information.

## 2.3 COLD START PROBLEM

The cold start problem is a common problem in real-world graph-structured data where new nodes may be added to an existing graph. Most work tackling the cold start problem focuses on solving the recommendation problem in a cold start setting. Recommendation systems are widely used in the real world applications such as friend recommendation, movie recommendation and purchase recommendation. In the recommendation problem, the task is to predict recommendation links from a heterogeneous graph containing two types of graphs; a user-user social graph and user-item rating graph. Users are involved in both types of graphs to connect the two types of graphs into a single heterogeneous graph. The collaborative filtering (CF) method is one of the traditional methods where a rating is predicted based on similarities between users or items. In short, the CF method makes recommendations based on ratings given by other users in the social graph. Sarwar et al. (2001) introduced an item-based CF method, which computes item-item similarities, and proved it outperforms other user-based collaborative filtering recommendation methods. Efforts to make use of CF for recommendation systems in the cold start scenario have been made. Lika et al. (2014) approached the cold user problem by exploiting CF followed by hard-clustering of users based on their attributes. Once the cluster of the new user has been decided, CF is only applied to members of the same cluster. Berg et al. (2017) formulated the matrix completion task of recommendation systems as a link prediction problem by viewing the user-item graph as a bipartite interaction graph. However, they instead used a warm start setting where at least one rating for a user is given which is not precisely a cold start assumption.

## 3 PROBLEM DEFINITION

A graph $G$ is denoted as a pair $(\mathcal{V}, \mathcal{E})$ with $\mathcal{V} = \{v_1, \cdots, v_N\}$ the set of nodes (vertices), and $\mathcal{E} \in \mathcal{V} \times \mathcal{V}$ the set of edges. Each node $v_i$ is associated with a feature vector $x_i \in \mathbb{R}^F$. To make notation more compact, the set of node feature vectors of graph $G$ is denoted as a matrix $X = [x_1, x_2, \cdots, x_N]^\top \in \mathbb{R}^{N \times F}$. Additionally, a graph has an $N$-by-$N$ adjacency matrix $A$ where $A_{i,j}$ represents the existence of an edge between $v_i$ and $v_j$ and a degree matrix $D$, a diagonal matrix which contains information about the degree of each node.

In the cold start setting, we will call those nodes that are newly introduced without any link from the existing graph *cold nodes* and let them be denoted by $v_i^c$. In the same manner, existing nodes in the graph are denoted as $v_j^e$. Then, $X_e$ only contains feature vectors of existing nodes $x_0^e, x_1^e, x_2^e, \ldots, x_{N_e}^e$ and $X_c$ contains feature vectors of cold nodes $x_0^c, x_1^c, x_2^c, \ldots, x_{N_c}^c$, where $N_c$ and $N_e$ are the number of existing nodes and cold nodes, respectively.

In our problem setting, we assume that new nodes are added to the existing graph and need to find possible links between new nodes and existing nodes followed by a node classification to figure out the hidden attributes of new nodes. Our problem definition is as follows: Given existing graph, $G = (V_e, A)$, and cold nodes $X_c$, find out all possible links between cold nodes $X_c$ and existing nodes $X_e$, then predict the class of cold nodes $X_c$.

## 4 METHOD

Here we present our method, the ColdExpand, that compensates the lack of topological information from the cold nodes by adding an additional task of semi-supervised node classification to a deep autoencoding principle in the stage of edge reconstruction.

### 4.1 LINK PREDICTION OF COLD NODES

Predicting possible edges of cold nodes is an important sub-goal for expanding a semi-supervised learning but is by itself still a difficult task because the only clue the new node provides is its own node features. As mentioned in Section 2.2, traditional similarity-based link prediction methods are not applicable in the cold start setting since no neighboring nodes of the cold node are known. On the other hand, learning-based link prediction algorithms (Passino et al., 2019; Kipf & Welling, 2016b; Pan et al., 2019) can be applied in the cold start setting by computing similarity scores between two embedded node feature vectors followed by a binary classification layer. Still, while embeddings of the existing nodes are conditioned on both the node feature and the features of neighboring nodes, embeddings of the cold nodes refer only to their own node features without any help from the topological structure of the graph resulting in isolated learning.

$$x_i^e = \text{GNN}_{encoder}(x_i^e, N(x_i^e)) \tag{1}$$

$$x_i^c = \text{GNN}_{encoder}(x_i^c, \{\varnothing\}) \tag{2}$$

when $N(x_i)$ is a set of neighboring nodes of node $x_i$ and weights of the $\text{GNN}_{encoder}$ are shared between the two. This hinders traditional GNN encoders from generalizing their training to the cold nodes. In order to make a common goal that can be accomplished by both existing nodes $x^e$ and cold nodes $x^c$, we train the GCN Encoder with an additional node classification goal as given by the following equation.

$$L_{total} = L_{link} + L_{classification} \tag{3}$$

### 4.2 COLDEXPAND

Figure 2 illustrates the overall structure of our ColdExpand model. Our model is composed of two parts; first, the GCN module for link prediction on the cold nodes and, secondly, the GCN module for classification on the cold nodes. We denote the first GCN module as GCN Encoder and the second GCN module as GCN Classifier. In the GCN Encoder, graph $G$ and a cold node $v^c$ are given as input. We use two basic GCN layers as suggested in (Kipf & Welling, 2016a) for an encoder to get the embedded node vector $Z$. In order to predict any possible links of the cold node $v^c$, we feed this embedded node vector $Z$ to an inner product decoder which is similar to previous link prediction

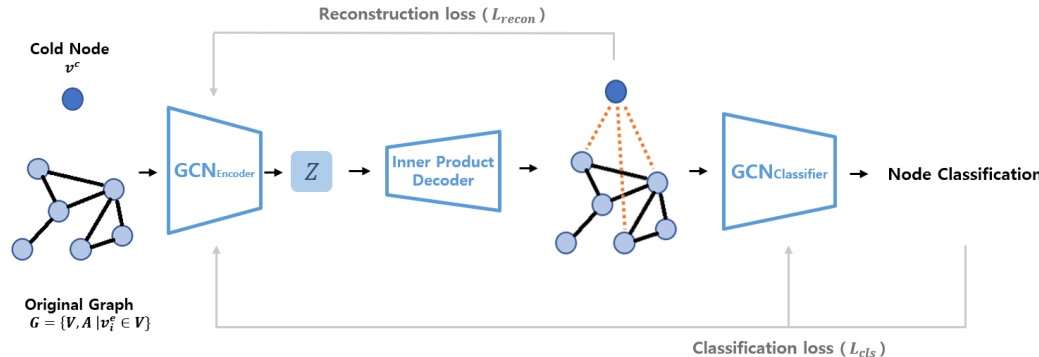

Figure 2: Computational graph for the ColdExpand model

methods (Passino et al., 2019; Kipf & Welling, 2016b; Pan et al., 2019). Along with original edges, predicted links are then fed to the GCN Classifier for node classification.

One GCN layer can be written as follows:

$$M(X, A, \Theta) = \tilde{D}^{-1/2} \tilde{A} \tilde{D}^{-1/2} X \Theta \tag{4}$$

where $\tilde{A}$ is an adjacency matrix of graph $G$ with additional self connections and $\tilde{D}$ is the degree matrix of $\tilde{A}$. $\Theta$ is a weight matrix for GCN module. Using the basic 2-layer GCN module, the embedding node matrix for all existing nodes $X_e$ and cold nodes $X_c$ is represented as follows:

$$q(Z_e|(X_e, A_{ee})) = M(\text{ReLU}(M(X_e, A_{ee}, \Theta_1), \Theta_2) \tag{5}$$
$$q(Z_c|(X_c, A_{ee})) = \text{ReLU}(X_c\Theta_1)\Theta_2 \tag{6}$$

where $\Theta_1$ is a weight matrix for the first-layer GCN and $\Theta_2$ for the second.

$$p(A_{ij}^{ec} = 1|z_i^e, z_j^c) = \sigma(z_i^e z_j^{c\top}) \tag{7}$$

The inner product decoder computes the sigmoid of the inner product between two embedded node vectors. $A_{ec}$ indicates possible edges between the existing nodes and the cold nodes. The probability of existence of a link between existing nodes $x_i^e$ and cold nodes $x_j^c$ is defined as follows:

Once the probabilities of links have been computed, new hard links for the cold nodes can be added.

$$A_{ec} = \begin{cases} 1, & \text{if } p(A_{ij}^{ec} = 1|z_i^e, z_j^c) > \xi \\ 0, & \text{otherwise} \end{cases} \tag{8}$$

where $\xi$ is a hyperparameter which is a threshold for hard links. During the experiment $\xi$ was set to 0.5.

$$\hat{A} = \begin{pmatrix} A_{ee} & A_{ec} \\ A_{ce} & A_{cc} \end{pmatrix} \tag{9}$$

$A_{cc}$ can be obtained in a same manner. Then the new $\hat{A}$ is fed to the GCN Classifier for node classification on the cold nodes. In order to maintain a continuous gradient along the model, we use the convolution operation with edge weights. Weights of all additional edges $A_{ec}$, $A_{cc}$ are determined by the probability score of the edge, $p(A_{ij}^{ec} = 1)$ while all the other weights for existing edges are fixed to 1.

$$\hat{Y} = \text{softmax}(\text{GCN}_{\text{Classifier}}(X, \hat{A})) \tag{10}$$

we only use $\hat{Y}_c \in \hat{Y}$ for class prediction on cold nodes. For optimization, we use two losses for multi-task learning. The first loss is a link reconstruction loss from the link prediction part of the model and the second loss is node classification loss from GCN Classifier.

$$L_{\text{recon}} = \mathbb{E}_{q(Z|(X,A))}[\log p(A_{ec}|Z) + \log p(A_{cc}|Z)] \tag{11}$$
$$L_{\text{cls}} = - \sum_{\hat{y}^c \in \hat{Y}_c} y^c \log \hat{y}^c \tag{12}$$
$$L_{total} = L_{\text{recon}} + \lambda L_{\text{cls}} \tag{13}$$

Finally, the total loss is represented as a weighted sum of the two losses. $\lambda$ is a hyperparameter for training.

Table 1: Dataset

| Name | Nodes | Edges | Classes | Cold Nodes |
|---|---|---|---|---|
| Cora | 3,327 | 4,732 | 6 | 332 |
| CiteSeer | 2,708 | 5,429 | 7 | 270 |
| PubMed | 19,717 | 44,338 | 3 | 1971 |

## 5 EXPERIMENT

### 5.1 DATASET

In order to evaluate the performance of graph learning in the cold start setting, we first select 10% of randomly shuffled nodes as a test set of cold nodes. Then we create positive edge samples of the set of cold nodes by parsing all links in and out of the cold nodes and every other non-links become negative samples. Whole positive and negative edge samples are used as test sets for the cold start link prediction task and are hidden in the training stage along with the class labels of cold nodes. This process is done in 10-fold cross-validation on all nodes in the dataset. In similar fashion, 5% of the nodes were masked as validation set and another 10% of the nodes as training set. During training we choose the same amount of negative edge samples as positive edge samples. At last, in the training period edges of 20% of nodes were masked. For semi-supervised node classification only 10% of the labels of the nodes were given.

### 5.2 BASELINE

Since there are not many prior works for cold start graph learning tasks, we created some baselines to prove the benefits of our model.

#### 5.2.1 EXPANDED SEMI-SUPERVISED LINK PREDICTION

We used learning-based link prediction algorithms GAE, VGAE (Kipf & Welling, 2016b) and ARGA (Pan et al., 2019) to compare the performance of link generation for cold nodes. For fair comparison we have restricted the number of edges used in training to 10%. Then each link prediction method was trained with the ColdExpand method by adding an additional task of node classification with loss function as given in Equation 13.

#### 5.2.2 EXPANDED SEMI-SUPERVISED NODE CLASSIFICATION

The simplest baseline for semi-supervised node classification is a linear model that only considers node attributes of the graph. The number of labels that the model can see was restricted for fair comparison.

$$E = \text{MLP}(X_e), \ \ \hat{y} = \text{softmax}(E) \tag{14}$$

The other baselines are vanilla GNNs which are trained on the existing graph and used to predict the labels of the cold nodes. In this setting, the same number of labels for existing nodes are provided during training.

$$E = \text{GNN}(X_e, A), \ \ \hat{y} = \text{softmax}(E) \tag{15}$$

There are several training strategies for our ColdExpand model. In the ColdExpand-Freeze model, first part of the model, the GCN Encoder, is pre-trained with a multi-task goal under fixed $\lambda$. Once pre-training is completed, weights of the GCN Encoder are frozen and training on the GCN Classifier takes place. Unlike the ColdExpand-Freeze model, the ColdExpand-Unfreeze model updates the weights of the GCN Encoder along with the GCN Classifier. For the ColdExpand models, we choose the best setting among several choices of $\lambda$. To demonstrate the advantage of adding an additional task, we also tested with ColdExpand-CLS which shares the same model architecture but only uses $L_{cls}$ for optimization. Each test was done with various convolutional GNN methods including GCN, GraphSAGE (Hamilton et al., 2017), and GAT (Veličković et al., 2017).

### 5.3 RESULTS

#### 5.3.1 RESULTS OF LINK PREDICTION ON COLD NODES

| Algorithm | Cora | CiteSeer | PubMed |
|---|---|---|---|
| GAE | $0.625 \pm 0.017$ | $0.678 \pm 0.029$ | $0.809 + 0.007$ |
| GAE (ColdExpand) | $\mathbf{0.813 \pm 0.021}$ | $\mathbf{0.854 \pm 0.019}$ | $\mathbf{0.907 \pm 0.006}$ |
| VGAE | $\mathbf{0.655 \pm 0.020}$ | $0.651 \pm 0.016$ | $0.779 \pm 0.030$ |
| VGAE (ColdExpand) | $0.643 \pm 0.034$ | $\mathbf{0.676 \pm 0.058}$ | $\mathbf{0.836 \pm 0.031}$ |
| ARGA | $0.639 \pm 0.018$ | $0.679 \pm 0.021$ | $0.858 \pm 0.016$ |
| ARGA (ColdExpand) | $\mathbf{0.784 \pm 0.024}$ | $\mathbf{0.824 \pm 0.024}$ | $\mathbf{0.881 \pm 0.006}$ |

Table 2: Results of link prediction on cold nodes in area under the ROC curve (AUC) scores. Bold score indicated the best AUC score. In every methods, adding our learning strategy generally increased performance.

Table 2 illustrates the results of expanded semi-supervised link prediction on cold nodes in *area under the ROC curve* (AUC) scores for each benchmark and dataset. Training with our ColdExpand strategy, ColdExpand encoders outperform most learning-based algorithms in reconstructing the links of the cold nodes. Especially in the Cora and CiteSeer datasets which have enough number of classes, our model improves the performance (at most 24%) by a large margin. In PubMed, where there are only 3 classes of labels, our model achieves 5.7% higher AUC score. With the additional training goal, our model definitely outperforms traditional link prediction methods on predicting the possible links of the cold nodes and training is more beneficial when there are enough number of classes.

#### 5.3.2 RESULTS OF SEMI-SUPERVISED NODE CLASSIFICATION ON COLD NODES

| Algorithm | Cora | CiteSeer | PubMed |
|---|---|---|---|
| **Linear Function** | $0.644 \pm 0.025$ | $0.650 \pm 0.021$ | $0.838 \pm 0.011$ |
| **GCN** | | | |
| Vanilla GCN | $0.690 \pm 0.023$ | $0.656 \pm 0.024$ | $0.836 \pm 0.006$ |
| ColdExpand-CLS | $0.445 \pm 0.153$ | $0.609 \pm 0.082$ | $0.768 \pm 0.028$ |
| ColdExpand-Freeze | $0.760 \pm 0.015$ | $0.688 \pm 0.026$ | $\mathbf{0.852 \pm 0.007}$ |
| ColdExpand-Unfreeze | $\mathbf{0.792 \pm 0.029}$ | $\mathbf{0.710 \pm 0.020}$ | $0.835 \pm 0.010$ |
| **GraphSAGE** | | | |
| Vanilla GraphSAGE | $0.545 \pm 0.058$ | $0.596 \pm 0.071$ | $0.801 \pm 0.011$ |
| ColdExpand-CLS | $0.444 \pm 0.148$ | $0.575 \pm 0.082$ | $0.771 \pm 0.017$ |
| ColdExpand-Freeze | $0.746 \pm 0.032$ | $0.683 \pm 0.023$ | $\mathbf{0.817 \pm 0.009}$ |
| ColdExpand-Unfreeze | $\mathbf{0.791 \pm 0.029}$ | $\mathbf{0.711 \pm 0.018}$ | $0.816 \pm 0.009$ |
| **GAT** | | | |
| Vanilla GAT | $0.534 \pm 0.040$ | $0.630 \pm 0.024$ | $0.796 \pm 0.016$ |
| ColdExpand-CLS | $0.490 \pm 0.126$ | $0.571 \pm 0.122$ | $0.764 \pm 0.022$ |
| ColdExpand-Freeze | $0.638 \pm 0.049$ | $0.634 \pm 0.030$ | $\mathbf{0.821 \pm 0.008}$ |
| ColdExpand-Unfreeze | $\mathbf{0.686 \pm 0.028}$ | $\mathbf{0.673 \pm 0.021}$ | $0.817 \pm 0.011$ |

Table 3: Experimental results on node classification task in comparison to various other models. Bold score indicated the best accuracy score. Our model outperformed existing GNN methods in every dataset.

Table 3 shows experimental results on the expanded semi-supervised node classification task in comparison to various other models. In order to test how inductive learning methods perform in the cold start setting, we tested with vanilla GCN, GraphSAGE, GAT methods by using the weights learned from non-cold start settings to classify cold nodes. On every dataset, our ColdExpand method performed better than traditional inductive GNN methods in the cold start setting. This is due to the

fundamental limitation that inductive learning methods have that they exclude cold nodes during training but rather test each node independently. In contrast, our ColdExpand model uses a multi-task strategy to incorporate cold nodes into the existing graph and furthermore include them in training. The linear function model which does not consider topological structure of a graph, generally performed poorly. However for PubMed, which has only three classes of labels, the linear function model showed performance close to our model. On other dataset, our ColdExpand model outperformed linear and vanilla GNN models by a big gap in performance: Cora (15%), CiteSeer (8%).

| Algorithm | Cora | CiteSeer | PubMed |
|---|---|---|---|
| Edges per node | 1.42 | 2.00 | 2.25 |
| ColdExpand-CLS | 5.96 | 6.30 | 8.23 |
| ColdExpand-Unfreeze | 5.12 | 4.63 | 6.73 |
| Decreased edges (%) | 14.1% | 26.5% | 18.3% |

Table 4: Average number of generated edges for each cold node by two different training strategies. Edges per node represents ground truth number of edges per each node. By using additional link prediction loss, ColdExpand-Unfreeze decreased at most 26.5% of new edges compared to ColdExpand-CLS which is closer to the ground truth amount.

Another notable result from the experiment was number of additional edges each GCN Encoder generates. In case of ColdExpand-CLS where classification loss was solely used for optimization, the number of the additional edges tends to explode, generating much more edges between cold nodes and existing nodes. On the other hand, other ColdExpand models generated relatively fewer edges (Table 4) using the link prediction loss as a regularizer for the GCN Encoder.

Results show that in general, ColdExpand-Unfreeze model works better than ColdExpand-Freeze model. While traditional link prediction methods focus on reconstructing 1-hop links, Klicpera et al. (2019) found that diffusion of graph, considering neighbors from multiple hops, improves semi-supervised learning without over-smoothing problem. This indicates that one-hop link prediction may not be the best solution for classifying the nodes. By training GCN Encoder during the stage of node classification we can find such links that can improve node classification that traditional link prediction methods could not find. In contrast, PubMed is a large graph with many nodes which degrades the effect of noisy connections.

### 5.3.3 ABLATION STUDY

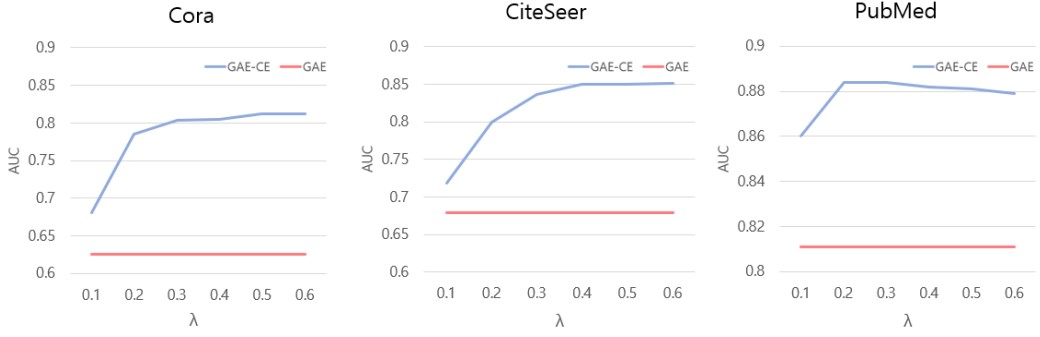

Figure 3: Link prediction results for different $\lambda$. GAE-CE represents results using ColdExpand strategy. We added results of GAE which is invariant to $\lambda$ for comparison.

Figure 3 illustrates trend of the results on cold start link prediction with varying hyperparameter, $\lambda$, which decides the amount of additional classification loss to be considered. Each dataset has different property resulting in a different optimal $\lambda$. From the trend we can easily figure out that

adding an additional task benefits cold start link prediction. This approach was more effective when there were enough distinct classes to map nodes to.

## 6 CONCLUSION

In this paper, we draw attention to a new task Expanded Semi-Supervised Learning which is a task to extend semi-supervised learning even to cold nodes. Even though the cold start environment is an extreme situation, most real-world applications must face it. We proposed the ColdExpand method which uses multi-task strategy to overcome the lack of topological information from cold nodes.

Our new task can be applied in many categories of machine learning. For example, community detection is a task of discovering overlapping groups of nodes, also referred to as communities, that share similar properties from the network graph. While most works on community detection only focused on static social networks (Leskovec & Mcauley, 2012; Chen et al., 2017; Li et al., 2019; Jia et al., 2019), it is critical that learning can be propagated even to new users. Our method can be used to categorize and adapt a new user to an existing community. In addition, solving the cold start problem in graphs is an important stepping stone for incremental learning. Incremental learning (Joshi & Kulkarni, 2012) is a learning policy that extends its existing model as new samples are dynamically added. Graphs that can grow incrementally can be used to broaden the knowledge in cognitive models that rely on topological structure such as Adaptive Resonance Theory (Carpenter & Grossberg, 2010) or the Neural Gas model (Fritzke, 1995).

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
