# OpenReview forum: "ColdExpand: Semi-Supervised Graph Learning in Cold Start"
_ICLR.cc/2021/Conference — Reject_

### Official Review · AnonReviewer2 · 2020-10-22
**ColdExpand**

**Rating:** 6
**Confidence:** 4

**Review:**

Summary:


This paper presents a model called ColdExpand that addresses learning tasks related to attributed graphs. Authors argue that this model is the first model able to deal with the cold-start problem, i.e. new nodes with no structure information, which explains its name. However I think they totally miss part of the recent literature on inductive approaches. ColdExpand addresses the new node issue by using an architecture that combines a deep auto-encoder and a GNN classifier. Experiments show that ColdExpand is better for both link prediction and node classification, but (again) it is not compared to more appropriate baselines.


##########################################################################

Reasons for score:

I vote for rejection because of two reasons. First, I think the cold-start problem presented in this paper can be solved by using inductive approaches that has already been developed. I guess that "inductive" can be understood in different ways, but previous works are able to deal with new nodes that are described by attributes only (see, for instance, G2G [1] and IDNE [2]).
Second, the model looks really incremental with a simple concatenation of AE and GNN. The semi-supervised task is not clearly described (it looks as if the whole dataset is labelled). As a minor remark, I find that evaluating the model on a unique data type (here, scientific networks) shed a bad light on the ability to generalize on diverse datasets.

[1] Aleksandar Bojchevski, Stephan Günnemann (2018), Deep Gaussian Embedding of Graphs: Unsupervised Inductive Learning via Ranking, ICLR 2018

[2] R. Brochier, A. Guille and J. Velcin (2020), Inductive Document Network Embedding with Topic-Word Attention, ECIR 2020

##########################################################################Pros:


1. The model addresses the more realistic task of inductive prediction, in contrast to transductive approaches.
2. The paper is reasonnably written.

##########################################################################

Cons:


1. Litterature on inductive approaches is missing (see above).
2. Model looks really incremental
3. It is not compared to inductive models


##########################################################################

Questions during rebuttal period:

Please clarify the novelty of the cold-start problem addressed in this paper and the usual inductive task as addressed in [1] and [2]. Besides I would like to know why it is a semi-supervised setting.


#########################################################################

I guess A should be mentioned in Eq. (5).

The authors should check the references (e.g., Velickovic et al. has been published at ICLR 2018).

Some typos:

"in a log of variant matrix"
"different hyperparameter"

######

Update: after reading the other reviews and author response, I decided to increase my grade to 6.

---

> ### Author Response · Authors · 2020-11-15
> **Response to Reviewer 2 (1/2)**
>
> First of all, we feel grateful to the reviewer for insightful and thorough feedback on our paper. These are our responses to your concerns.
>
> 1. Can’t Inductive learning be a solution for Expanded Semi-supervised Learning?
>
> We did not completely ignore the use of inductive methods in our expanded semi-supervised learning tasks. Inductive method learns rules from known samples to predict unknown samples. Inductive methods such as GCN, GraphSAGE, GAT can still classify cold nodes using the weights learned from non-cold node settings. Corresponding results can be seen at Vanilla GCN, GraphSAGE, GAT methods in Table 3 of our paper. However we can see that the results are almost same or less than results of that using basic Linear Functions. It is because of the limitation that inductive learning has, that it excludes cold nodes during the training. We will explain more of it below.
>
> G2G method, that the reviewer referenced, is a method that assumes each node as a gaussian distribution and uses KL divergence between two nodes’ distribution to train mean function and variance function so that the distance between the two distributions can be proportional to a shortest path length between two nodes. In this case G2G learns to map node attributes to an embedding space without any help from it’s neighbor. However, the G2G method is still an inductive method which has the same limitation to be applied to our problem as the above methods.
>
> Using an inductive method to generalize learning to the cold node is different from solving Expanded Semi-supervised learning. Semi-supervised learning is a learning method that uses unseen data to approximate underlying marginal data distribution p(x) over the input space to better approximate on posterior distribution p(y|x) [1]. However, inductive learning of G2G does not include cold nodes(unseen target nodes) during the training but just uses a pre-trained model to test those. In short, G2G can not use additional cold nodes or their node attribute data to better understand the underlying marginal data distribution of a graph. This is why the Vanilla inductive method showed similar accuracy with the linear functions(same as a graph with no edge).
>
> In contrast, our ColdExpand model uses multi-task strategy to include cold nodes to an existing graph and further to the training. For a short demonstration of the difference between inductive learning and our ColdExpand, when 100 cold nodes appear, the G2G model will test all 100 nodes independently. However our model not only builds connections between cold nodes and existing nodes(A_ec) but also between cold nodes and cold nodes(A_cc) and uses expanded graph for training. To this end, while our model can be a solution for semi-supervised learning on cold nodes, inductive learning can not be one. We appreciate your comments, the discussion and reference will be added in the final manuscript.
>
>
> [1] Jesper E Van Engelen and Holger H Hoos. A survey on semi-supervised learning. Machine Learning, 109(2):373–440, 2020.

---

> > ### Author Response · Authors · 2020-11-15
> > **Response to Reviewer 2 (2/2)**
> >
> > 2.  Semi-supervised Setting
> >
> > Sorry for the inconvenience. It is  mentioned in the last sentence of Section 5.1 Dataset, that we gave labels for 10% of nodes for each graph data to make it a semi-supervised setting.
> >
> > 3.   Model is simple concatenation of two models
> >
> > We think our novelty comes from the overlooked problem we suggest and the multi-task strategy to overcome an extreme situation with little information. We started from the assumption that link prediction task and node classification task in fact share a common goal which is mapping similar nodes to similar space. Most edges represent similarity between nodes and similar nodes tend to be in the same class. There can be various models using multi-task strategy and our ColdExpand is a simple and valid one which sufficiently improves existing methods to face cold start problem in Expanded Semi-supervised learning tasks.
> > As also mentioned by the Reviewer 4, our contribution is to point out the overlooked real-world problem of expanded semi-supervised tasks in cold start setting and provide a simple but powerful method that tackles the problem.
> >
> >
> > Again we thank the reviewer for the thoughtful review. Hope our rebuttal resolves the concerns.

---

> > > ### Comment · AnonReviewer2 · 2020-11-16
> > > **incremental**
> > >
> > > Ok my mistake, I miss the information for the percentage.
> > >
> > > The second point has been highlighted by two other reviewers (rev1 and rev3). It's clear that most of science is incremental ("...on the shoulder of giants"). However, I expected a clearer demonstration that the new combination is better than previous approaches.

---

> > > > ### Author Response · Authors · 2020-11-18
> > > > **2nd Response to Reviewer 2**
> > > >
> > > > Thank you for the comment. Though we have to point out some points that we think you might have misunderstood.
> > > >
> > > > 1. Our model is not a supervised model
> > > >
> > > > During both tasks of link prediction and node classification our model uses other unlabeled nodes which make our model a solid semi-supervised method. As you said, it also can be tackled with a supervised approach, but clearly, it is more beneficial to use other structural information of the given graphs (Table 3).
> > > >
> > > >
> > > > 2. We never said that our model is the first one able to deal with cold start problem.
> > > >
> > > > What we wrote in the paper is that our model is the first one to enable semi-supervised learning on the cold nodes (This is different from JUST classifying the cold nodes as mentioned in the demonstration of the previous rebuttal). There is a big difference between the two statements because our model can use extra information from cold nodes to expand semi-supervised learning which no other models have done.
> > > >
> > > >
> > > > 3. GraphSAGE works poorly in cold start settings.
> > > >
> > > > As mentioned in the rebuttal for Reviewer3, vanilla GraphSAGE operates semi-supervised node classification on test nodes with access to their neighboring node which cannot happen in cold start setting because of the lack of topological information. This is why GraphSAGE shows low performance when tested in cold start situations without any neighboring nodes and we did corresponding experiments to prove it (Table 3).
> > > >
> > > > 4. Clearer demonstration that the new combination is better than previous approaches.
> > > >
> > > > Our method is, until now, the only method that can take advantage of unseen cold nodes to solve semi-supervised tasks on cold nodes while previous models only focus on solving one task on each cold node independently.
> > > >
> > > > We again thank you for your feedback. If there are any unclarified aspects, please let us know.

---

> > > > > ### Comment · AnonReviewer2 · 2020-11-18
> > > > > **short clarification**
> > > > >
> > > > > Thank you for taking the time to answer.
> > > > >
> > > > > When I wrote that your are "on the supervised face of the coin", I don't mean that your task is supervised classification with examples all labelled. Semi-supervised learning can be understood in two ways. First, the final task can be to predict a value or a class ("supervised view"). Second, the final task can be clustering ("unsupervised view"). The latter is when you add some kind of supervision, such as must-link cannot-link constraints.

---

> > ### Comment · AnonReviewer2 · 2020-11-16
> > **ok for semi-supervised but not enough**
> >
> > Thank you for clarifying: your approach is clearly on the supervised face of the coin: the overall framework is dedicated to predict classes, and it uses additional unlabelled data. I must say that the paper wasn't fully clear, which can explain my main concern related to inductive vs. transductive issues.
> >
> > Therefore GNG wasn't the best example I could take. However, as written by Rev3, GraphSAGE is a better example of an inductive solution and we expect you to prove the interest of your approach in comparison to such sota approach.
> >
> > All in all, I'm still leaning toward rejection. I disagree with your statement that your model is the first one able to deal with the cold-start problem.

---

### Official Review · AnonReviewer3 · 2020-10-28
**Recommendation for Rejection (title before rebuttal phase)**

**Rating:** 6
**Confidence:** 5

**Review:**

Summary
========
This paper is about the cold-start problem for representation learning on dynamics graphs. More specifically, the proposed method (ColdExpand) uses convolutional networks and multi-task learning (node classification loss and link prediction loss) to learn embeddings for new unseen nodes in the graph, i.e., nodes that we only know their features and not their connectivity to the existing graph. Such a problem is useful for applications of the link prediction problem and more specifically, recommendation tasks or graph completion tasks. The authors test their technique using three benchmark datasets (Cora, CiteSeer and PubMed) and by using building blocks of existing works they create some baselines to compare. They claim to be the first study for semi-supervised learning to unseen nodes. However, there are a couple of works in this field already, as for example the GraphSAGE model that they cite in this paper (see below for more).

Reasons for score
================
Overall, the paper is about an interesting problem (cold-start for representation learning on dynamic graphs). However, the proposed methods has limited novelty by using known building blocks from the literature. Also, the experimental setup and the results are not very convincing. Results on more datasets, more comparison methods and a different setup will strengthen the paper.


Strengths
========
- The problem of representation learning in dynamic graphs for unseen nodes is interesting and has attracted the attention of researchers in the field in the last years. In this paper, the authors propose a semi-supervised learning for the link prediction for unseen nodes.
- The proposed method that uses a multi-task learning strategy by combining a loss for link reconstruction and a loss for node classification is reasonable, has been well motivated in the paper, despite the limited novelty.
- Overall the paper is well-written and has a nice flow.


Weaknesses
===========
- The authors claim that this is the first work on semi-supervised learning for unseen nodes. However, there are a couple of methods already in this field. Other methods include: GraphSAGE (already in the references of the paper), PinSAGE [1], HAN [2] and DynRep [3]. The authors should consider to add these methods in the comparison, and also in related work subsection 2.3.
- Figure 1 is not useful and takes a lot of space, the authors could use the space to expand their experiments.
- In subsection 4.2 there are some parameters that are never described, as for example \Theta.  In equation 4, M(,,) takes 3 arguments, but later it is used with only two. In general there are some inconsistencies in the notation and the equations in terms of the definition/description of some of the parameters. It needs careful check.
- The experimentation is limited. The authors use three datasets and they apply some preprocessing in order to use some nodes/edges as unseen. It would be nice if the authors can add these numbers in Table 1.
- It would be interesting to see results for the GCN, GraphSAGE and GAT in the link prediction task. An aggregation of the node embeddings from the nodes that create an edge can be used to compute an edge embedding (as it is already done in the literature).
- It could be interesting to add an experiment with a more realistic imbalance in the dataset (more negative edges, and less positive ones).
- In Figure 3, it is not obvious what the light blue and dark blue lines represent. I would suggest to add the x-axis, y-axis and labels in the figure and remove them from the caption if there is no space.
- Overall, more datasets (e.g., Reddit, PPI) and more comparison methods (e.g., GraphSAGE (for link prediction), HAN, DynRep) will make the results more convincing and insightful.
- There is no comment on the scalability/runtime of the proposed method.
- The results in some cases are very close, it would be better if the authors could test for statistical significance and report the scores.


[1] Ying, R., He, R., Chen, K., Eksombatchai, P., Hamilton, W.L. and Leskovec, J., 2018, July. Graph convolutional neural networks for web-scale recommender systems. In Proceedings of the 24th ACM SIGKDD International Conference on Knowledge Discovery & Data Mining (pp. 974-983).

[2] Xiao Wang, Houye Ji, Chuan Shi, Bai Wang, Yanfang Ye, Peng Cui, and Philip S Yu. 2019. Heterogeneous Graph Attention Network. In <i>The World Wide Web Conference</i> (<i>WWW '19</i>). Association for Computing Machinery, New York, NY, USA, 2022–2032. DOI:https://doi.org/10.1145/3308558.3313562

[3] Trivedi, Rakshit, Farajtabar, Mehrdad, Biswal, Prasenjeet, and Zha, Hongyuan.. "DyRep: Learning Representations over Dynamic Graphs". International Conference on Learning Representations (). Country unknown/Code not available. https://par.nsf.gov/biblio/10099025.


Questions during rebuttal
=====================
- Please address and clarify the suggestions mentioned in the weaknesses above.

---

> ### Author Response · Authors · 2020-11-16
> **Response to Reviewer 3**
>
> First of all, we feel grateful to the reviewer for insightful  and thorough feedback on our paper. These are our responses to your concerns.
>
> 1. There are already some methods for semi-supervised learning on unseen nodes.
>
> As the reviewer mentioned, in this paper we define “unseen” nodes as extreme cold nodes without ANY topological information, while most semi-supervised learning methods, including those the reviewer referred to, test for unseen nodes using information from their neighboring nodes. To our best knowledge, our method is still the first work to expand semi-supervised learning to unseen cold nodes.
>
> GraphSAGE mentions semi-supervised node classification on unseen nodes but it is different from our definition of “unseen”, because they access neighboring nodes of unseen (test) nodes in test time. For this reason, GraphSAGE shows low performance when tested in cold start situations without any neighboring nodes and we did corresponding experiments to prove it (Table 3). GraphSAGE also proposes another task, multi-graph generalization task, that learns from multiple training graphs to predict on unseen graphs provided by PPI dataset but it is far apart from our setting.
>
> It is the same for PinSage where they select T most influential neighbors(decided by random walks) of unseen nodes during the test for embedding. DynRep considers a dynamic graph where new nodes appear to the original graph, but it is different from our setting because the edges of the new nodes are pre-decided by the given evolving scenario.
>
> For these reasons, our method is distinct from the referred methods in that we tackle expanded semi-supervised learning for cold nodes without any help from their neighboring nodes. We appreciate the comment from the reviewer and will clarify the differences between such methods to ours in the final manuscript.
>
> 2. Proposed model has limited novelty
>
> We think our novelty comes from pointing out the overlooked problem and suggesting the simple model which uses multi-task strategy that can overcome an extreme cold start situation. There can be various models using multi-task strategy and our ColdExpand is a simple and valid one which sufficiently improves existing methods to face cold start problem in Expanded Semi-supervised learning tasks.
>
> 3. Need to strengthen the experiment
>
> Here are link prediction results for GraphSAGE and GAT compared to using GCN for link reconstruction.
>
> |           |Cora                |Citeseer         |PubMed         |
>
> |GCN   |0.625 ± 0.017 |0.678 ± 0.029 |0.809 ± 0.007|
>
> |SAGE  |0.613 ± 0.030 |0.687 ± 0.030 |0.700 ± 0.105|
>
> |GAT    |0.614 ± 0.026 |0.681 ± 0.028 |0.768 ± 0.009|
>
> We appreciate your comment and will add this additional experimental results in the final version of the paper. In addition, according to the reviewer's suggestion we are currently adding a Reddit dataset to our benchmark.
>
> 4. More information in Table 1
>
> We will give additional information for each dataset including the mean degree of each node, total number of cold nodes and total number of given labels.
>
> 5. Correction in Figure 1
>
> We think showing graphics for new task, Expanded Semi-supervised learning, is important. We are making a new figure that can be more intuitive and compact.
>
> Again we thank the reviewer for the thoughtful review. Hope our rebuttal resolves the concerns.

---

### Official Review · AnonReviewer4 · 2020-10-29
**Using link-prediction/retrieval to generate hard links in cold-start node classification is effective.**

**Rating:** 9
**Confidence:** 5

**Review:**

##########################################################################
Summary:

The paper proposes a new task in Graph Learning. Basically, the idea is the following: suppose we have a node classification model trained on a Graph G, suppose we have a new node (not present in G) and we want to classify it. Given that the new node has no connections with G’s other nodes we cannot leverage any structural information to run the classifier. This is an issue that authors present it as cold-start in semi-supervised graph learning. The solution, even if simple, is very effective and for this reason even more interesting. Basically, they start with a retrieval step (they call it link prediction) that is trained using a link reconstruction loss and it’s based on dot-product to make the link prediction phase scalable. After those links are reconstructed they run a node classification step (based on GCN, GraphSAGE, or GAT) on G plus the new node and the predicted links.  The results obtained in the experiments are really encouraging with improvements ranging from 15 to 25% over a baseline that does not consider the link prediction phase.
##########################################################################
Reasons for score:

Overall, I vote for accepting. I consider this as a non-trivial step forward towards using Graph Learning on recommendation problems and node classification problems. There are many applications of Graph Learning where the technique presented in this paper can be of help. Consider, for instance, all the words relying on GNNs to do fake news detection based on a semi-supervised technique. All of those methods fail in the case a new document (node) has no explicit connections to the graph. This method would solve that issue.

##########################################################################Pros:

1. The paper tackles a problem that is, in my opinion, very important and, so far, overlooked: cold-start node prediction using graph learning.
2. The technique presented in the paper is simple, which I consider a plus. It works and it’s simple.
3. The experiments show a great improvement over the baseline

##########################################################################
Cons:

1. One big limitation of this work, that is in my opinion under explored is that it is based on the assumption that links depend on the content/features of the nodes. In some cases this assumption might not hold true. I would like authors to discuss on this point.
2. I am not sure how the link prediction phase could be made scalable. As it is defined now it is an O(n^2) step. Or better, I have some ideas but I’d like authors to discuss this.
3. Why you pick a threshold of .5 in equation (8) Shouldn’t this be an hyperparam?

##########################################################################
Questions during rebuttal period:

Please address and clarify the cons above

#########################################################################
Some minor issues
(1) There is an inconsistency in the notation for M in equation (4) and M in equation (5). In equation (4) M takes 3 parameters, while in (5) it takes 2. Please clarify.
(2) Why you speak about cold nodes X_c instead of  cold node? Up until now it looked like you only used one node at a time. Please clarify.
(3) What is q() in equations (5) and (6)?
(4) In Figure 3, what is the black line under the blue curve? It’s not written anywhere.

---

> ### Author Response · Authors · 2020-11-16
> **Response to Reviewer 4**
>
> First of all, we feel grateful to the reviewer for insightful  and valuable feedback on our paper. These are our responses to your concerns.
>
> 1. Assumption that links depend on the content/features of the nodes
>
> We agree with the reviewer that in some cases, links may not be related to the feature of the nodes but to other aspects. For instance, visual scene graph[1] represents topological relations among objects in the scene that are not based on similarity between the objects but rather on predicates or location. Fortunately, in most cases we encounter, links in the graph represent similarity between two node attributes as many previous works assumed and our work also keeps the scope to such graphs. We will mention this assumption clearly in our final manuscript.
>
> [1] Krishna, Ranjay, et al. "Visual genome: Connecting language and vision using crowdsourced dense image annotations." International journal of computer vision 123.1 (2017): 32-73.
>
> 2. Link Prediction phase might not be scalable.
>
> Basically, the link prediction phase is bounded by O(N^2) time complexity to inference every link pair. However, once the training is done, we only have to generate edges from the cold nodes and the complexity of the link prediction phase now bounds to O(NC) when N is a number of existing nodes and C as a number of cold nodes(C<<N).
>
> Though, when we have to deal with extremely large graphs such as social networks, it will be wasteful to global search all possible edges from the existing graph. In such a case, we can cluster the graph to create representative super-nodes from each cluster and decide which super-node(cluster) is most likely to have edges with a cold node. Then we can efficiently search nodes in the cluster for possible links connected to the cold node.
>
> On top of it, if you have other ideas to reduce this complexity, we’d like to discuss it with you.
>
> 3. Shouldn’t 0.5 be hyperparameter?
>
> Yes, that is true. 0.5 is one possible hyperparameter for the threshold that decides the hard boundary between hard links and no-links. We fixed this value for easy evaluation on ablation results of different hyperparameter \lambda. We will clarify this part in the final manuscript.
>
> 4. Clarification on  X_c.
>
> X_c is a plural cold nodes because our method can consider a batch of cold nodes. Our method can also create edges between the cold nodes to better approximate the marginal distribution of the data p(x).
>
> 5. What is q() function?
>
> q function is an encoder function.
>
> 6. Clarification on Figure 3.
>
> We apologize for the confusion. The light blue line on the top represents different link prediction results of GAE-ColdExpand on different hyperparameter \lambda. The dark blue line on the bottom represents the baseline result of the original GAE which is invariant to change of \lambda. We will make sure that the explanation of Figure 3 be clear in the final manuscript.
>
> Again we thank the reviewer for the thoughtful review. Hope our rebuttal resolves the concerns.

---

### Official Review · AnonReviewer1 · 2020-11-10
**Lack of novelty, experiments should be improved.**

**Rating:** 5
**Confidence:** 3

**Review:**

Clarity : Eqn(5) and (6) introduce symbols $\theta_1$ and $\theta_2$ whose meaning is unclear. Eqn(6) requires more clarity as it is important to understand how intermediate embeddings for cold nodes are being generated.
Overall the problem motivation and related work section is well described

Novelty:  The proposed model is created by clubbing together GCN model for node classification and GAE model  for link prediction. The loss function is a weighted combination of the two. Lacks sufficient novelty
Impact: The paper clubs together two earlier well known models and makes no additional theoretical contributions.

Correctness: The link recommendation test sets have the same no. of positive and negative ratio. This scenario is highly unlikely in a real-world setting. A more skewed dataset would throw off the performance metrics significantly.
AUC scores are reported. AP/RR/Accuracy values should be explored as well. While I agree AUC is well prevalent in community, Precision captures the sensitivity in ranking much better in link prediction.

Maybe I missed the code, can the reviewer point out to it?

---

> ### Author Response · Authors · 2020-11-16
> **Response to Reviewer 1**
>
> First of all, we feel grateful to the reviewer for insightful  and thorough feedback on our paper. These are our responses to your concerns.
>
> 1. Equal number of pos, neg samples unlikely in a real-world setting.
>
> As we mentioned in the public comment, we used every single negative edge for link prediction test. We only used an equal number of positive and negative samples for the training and for our testing results on the table we tested on positive edges and every single negative edges which is highly unbalanced.
>
> 2. Clarification for Eq 5 and 6.
>
> Theta 1, 2 in Eq 5 and 6 indicates the weight matrix from each GCN layer 1 and 2. Eq 6 is a process of matmul between the weight matrix of GCN layer and cold node feature matrix since cold nodes do not have any neighboring nodes to consider.
>
> 3. Lack of novelty of the model.
>
> We think our novelty comes from pointing out the overlooked problem and suggesting the simple model which uses multi-task strategy that can overcome an extreme cold start situation. There can be various models using multi-task strategy and our ColdExpand is a simple and valid one which sufficiently improves existing methods to face cold start problem in Expanded Semi-supervised learning tasks.
>
> 4. Strengthen on the experiment in the link prediction part.
>
> We thought AUC score sufficiently expresses the advantage of our method but we can definitely add other metrics for link prediction method. We will make sure that it will be shown in the final manuscript.
>
> Again we thank the reviewer for the thoughtful review. Hope our rebuttal resolves the concerns.

---

### Author Response · Authors · 2020-11-12
**Correction made in experimental setup part**

Dear all,
We made a correction where we mis-wrote the part about making the test pair for the link prediction task. We wrote
“We also create the same number of randomly selected negative edge samples for each cold nodes for competitive training. Positive and negative edge samples are used as test sets for the cold start link prediction task and are hidden in the training stage along with the class labels of cold nodes. “
However we only used an equal number of positive and negative samples for the training and for our testing results on the table we tested on positive edges and every single negative edges which is highly unbalanced.

---

> ### Author Response · Authors · 2020-11-25
> **Revision Updated**
>
> Dear reviewers,
> we thank you for your thoughtful feedback on our paper. We updated our paper along with some corrections that reviewers required.
>
> (1) We made some fixation on Figure 1 along with it's caption.
>
> (2) Some equations on Section 4.2 have been fixed. Also we strengthened explanations on several parameters.
>
> (3) We added the number of Cold Nodes column in Table 1.
>
> (4) We added some discussions on the results of inductive methods in Section 5.3.2
>
> (5) Axis labels of Figure 3 were added with precise explanation on what each line indicates.
>
> (6) Additional experimental results on number of edges generated were added (Table 4)

---

### Decision · Program_Chairs · 2021-01-07
**Final Decision**

**Decision:**

Reject

**Comment:**

The reviewers agree that the paper is addressing an interesting problem (cold-start for representation learning on dynamic graphs). However, the proposed methods can be improved by proposing more novel ideas. At the moment, the proposed methods is a combination of GCN model for node classification and GAE model for link prediction. In this case, some analysis or theoretical justification may make the paper more interesting. Furthermore, the reviewers think the experiments can be improved. For instance, results on more datasets, more comparison methods and a different setup will strengthen the paper.